# TYK2 in Tumor Immunosurveillance

**DOI:** 10.3390/cancers12010150

**Published:** 2020-01-08

**Authors:** Anzhelika Karjalainen, Stephen Shoebridge, Milica Krunic, Natalija Simonović, Graham Tebb, Sabine Macho-Maschler, Birgit Strobl, Mathias Müller

**Affiliations:** Institute of Animal Breeding and Genetics, University of Veterinary Medicine Vienna, A-1210 Vienna, Austria; Anzhelika.karjalainen@vetmeduni.ac.at (A.K.); Stephen.Shoebridge@vetmeduni.ac.at (S.S.); milica.krunic@vetmeduni.ac.at (M.K.); sabine.macho-maschler@vetmeduni.ac.at (S.M.-M.)

**Keywords:** tyrosine kinase 2, Janus kinase (JAK) family, STATs, immunoediting, cancer-immunity cycle, interferons, interleukins

## Abstract

We review the history of the tyrosine kinase 2 (TYK2) as the founding member of the Janus kinase (JAK) family and outline its structure-function relation. Gene-targeted mice and hereditary defects of TYK2 in men have established the biological and pathological functions of TYK2 in innate and adaptive immune responses to infection and cancer and in (auto-)inflammation. We describe the architecture of the main cytokine receptor families associated with TYK2, which activate signal transducers and activators of transcription (STATs). We summarize the cytokine receptor activities with well characterized dependency on TYK2, the types of cells that respond to cytokines and TYK2 signaling-induced cytokine production. TYK2 may drive beneficial or detrimental activities, which we explain based on the concepts of tumor immunoediting and the cancer-immunity cycle in the tumor microenvironment. Finally, we summarize current knowledge of TYK2 functions in mouse models of tumor surveillance. The biology and biochemistry of JAKs, TYK2-dependent cytokines and cytokine signaling in tumor surveillance are well covered in recent reviews and the oncogenic properties of TYK2 are reviewed in the recent Special Issue ‘Targeting STAT3 and STAT5 in Cancer’ of Cancers.

## 1. Tumor Immunosurveillance and Tumor Microenvironment

The tumor immunosurveillance hypothesis, its history and its validation by elegant experiments in cancer mouse models have been reviewed recently [1,2,3]. Tumor surveillance is the means by which the immune system identifies and eliminates (pro-)tumorous cells based on their expression of stress-induced molecules or tumor-specific antigens. The ability of cancers to develop despite an intact host immune system has long puzzled biologists. A recent explanation relies on the concept of tumor immunoediting, which takes place in three phases: elimination, equilibrium and escape [4,5,6]. The elimination phase encompasses the processes originally classified as immunosurveillance. If only a proportion of transformed cells are eliminated, there will be a temporary equilibrium between the developing tumor and the protective function of the immune system. However, most tumors eventually escape: continuous changes in chromatin activities and/or the accumulation of DNA mutations in tumor cells lead to the appearance of tumor cell variants able to suppress, avoid or resist the antitumor immune responses [7,8]. The immunoselection of tumor variants lacking strong tumor-specific antigens by CD8^+^ and CD4^+^ T cells [9,10] represents another mechanism by which cancer cells escape tumor immunity.

Clinical outcome studies in human cancers have confirmed the validity of the immunoediting concept and have shown that the failure of immune protection in many patients is caused by negative regulators of T-cell responses in lymphoid organs (checkpoints) and in the tumor microenvironment (TME). This finding has led to a complementary concept, the cancer-immunity cycle, which recognizes that the immune response in cancer is stepwise and regulated by feedforward and feedback loops, ultimately leading to the elimination of cancer cells or the perpetuation of carcinogenesis by the induction of factors that impair tumor immunity [11,12].

Inflammation in the TME is a hallmark of cancer and counteracts immune surveillance and responses to therapy [13]. The tissues, cells and noncellular components that are involved in immunity to cancer and that constitute the TMEs of solid and hematopoietic cancers have been reviewed recently [14,15,16,17,18,19]. The balance between tumor-promoting inflammation and anticancer immunity in the TME is governed by pro-inflammatory and effector cytokines. Depending on the context and duration of activity, a single cytokine may exert anti- or pro-tumorigenic activities [20,21].

Tyrosine kinase 2 (TYK2) propagates the signals of cytokines that shape the TME. Type I and type II interferons (IFNs) are pivotal in anticancer immunity [22,23]. TYK2 is central for the response to type I IFN and for the production of type II IFN as well as for the response to the IL-12 and IL-10 family of cytokines, which control immunity and inflammation (see below). The pleiotropic and context-dependent activities of these cytokines show that TYK2 is responsible for tipping the balance between tumor-restrictive and tumor-promoting functions.

## 2. Identification and Structure-Function Relations of TYK2

TYK2 was the first member of the family of Janus kinases (JAK1-3, TYK2) to be cloned [24,25,26]. It was also the first JAK for which a biological function was demonstrated by genetic complementation of an IFN-unresponsive human cell line generated by saturation mutagenesis and selection against type I IFN responsiveness [27]. The basic biological functions of the three other JAKs have since been elucidated by work with IFN-unresponsive cell lines, gene-targeted JAK-deficient mice and human patients with inherited JAK deficiencies [28,29,30,31,32].

The structural features of TYK2 have been reviewed elsewhere [33,34,35,36,37]. Together with JAK1 or JAK2, TYK2 is associated with heterodimeric cytokine receptor chains (see Table 1 and below). Association of JAKs with their receptor generally stabilizes the anchoring at the cell surface and regulates the endocytic trafficking of cytokine receptors [38,39]. The mechanism is best understood for TYK2 at the type I IFN receptor (IFNAR) in human cells [40,41] and is similar for other cytokine/receptor combinations [42,43]. The conformational changes of ligand-bound cytokine receptors release the autoinhibitory intramolecular fold of the kinase domain [44,45,46]. In the case of TYK2, this leads to enzymatic activation by phosphorylation of two conserved tyrosine residues in the activation loop (Y1054/Y1055 in men and Y1047/Y1048 in mice) by transactivation of the adjacent JAK1/2 or by autophosphorylation. Cytokine-activated JAKs phosphorylate several tyrosine residues on the cytoplasmic receptor chains, creating docking sites for the SH2 domains of members of the signal transducer and activator of transcription (STAT) family. STATs are phosphorylated on a critical C-terminal tyrosine, undergo conformational changes, translocate as homo- or heterodimers to the nucleus and regulate transcriptional activity [32,47,48]. Some STATs also exhibit transcriptional activity when non-phosphorylated; these have been covered in recent reviews [49,50]. TYK2 activity is negatively regulated by intrinsic mechanisms, such as post-translational modifications and conformational inhibition [33,37], and by extrinsic effectors, such as protein tyrosine phosphatases (PTPs, e.g., PTB1B, SHP1 [51,52]) and suppressor of cytokine signaling (SOCS) proteins (e.g., SOCS1, SOCS3 [53]). In addition, chaperones, in particular heat shock protein (HSP) 90 [54], and noncoding RNAs [55,56] influence the stability and production of TYK2.

## 3. Loss-of-Function Mutations of TYK2 in Men and Mice

*TYK2/Tyk2* is located on chromosome 19 in men and on chromosome 9 in mice. The gene is expressed in all tissues. The first comprehensive studies of the biology of TYK2 relied on genetically engineered mice with a *Tyk2* loss-of-function (LOF) [64,65,66] or on the tissue-specific ablation of TYK2 [67]. They established that TYK2 is required for the immune response to infections and cancer and for the development of inflammation. Additionally, spontaneously mutated mouse strains have been discovered that carry mutations causing TYK2 deficiency: the B10.D1-H2^q^/SgJ mouse strain harbors an amino acid exchange (TYK2^E775K^) that destabilizes the protein [68], while the SWR/J or SJL/J strains harbor *Tyk2* promoter mutations that reduce TYK2 levels to below the limit of biochemical detection [69]. *TYK2* promoter variants in men are associated with an increased risk of virus-induced diabetes [70,71]. The first report of an inborn missense mutation leading to loss of TYK2 in a human patient stems from 2006; since then a further nine patients with complete loss of TYK2 have been reported [42,43,72,73]. Despite the fundamental differences in environmental conditions between men and mice and the extensive medical interventions in human patients, the effects of TYK2 deficiency in the two species are highly similar in terms of cytokine responses (see below and Table 2). In addition to its enzymatic activity, TYK2 has scaffolding functions, e.g., in endocytic cytokine receptor trafficking, PI3K signaling crosstalk and basal mitochondrial respiration [49,52]. To study the kinase-independent functions of TYK2 *in vivo* and to evaluate the efficacy and side effects of pharmaceutical inhibitors, we and others have generated mice expressing TYK2 with a point mutation in the ATP-binding pocket of the kinase domain (*Tyk2^K923E^* and *Tyk2^K923R-TG-Cre^*) [74,75]. *Tyk2^K923E^* mice phenocopy mice lacking TYK2 with respect to cytokine responses and susceptibility to viral infections [74,76]. In men, a common and a less frequent missense allele lead to expression of TYK2^P1104A^ or TYK2^I684S^, which lack catalytic activity and impair cytokine responses [77,78,79,80,81,82,83,84]. T*yk2^P1124A^* mice carrying the corresponding amino acid substitutions recapitulate the phenotypes in men [79,82].

## 4. TYK2-Dependent Cytokine Responses and Their Involvement in Immunity to Cancer

TYK2 is associated with heterodimeric/multimeric cytokine receptors, where it acts in concert with JAK1 or JAK2. Comprehensive reviews list the cytokines and growth factors able to activate TYK2 by phosphorylation of the critical tyrosine residues in the activation loop [35,104,105,106]. Here we focus on the cytokine-TYK2 signaling that transduces TYK2 phosphorylation into physiological changes validated by TYK2 deficiency in patients or mouse models (see Table 2) and/or by pharmacological inhibition of TYK2 [107,108,109,110]. Signaling molecules include the interleukin (IL)-12 family (IL-12, IL-23), the type I IFN subfamily (including IFNα subtypes, IFNβ), and the IL-10 family (IL-10, IL-22, IL-26, type III IFNs) [37,111,112]. Some TYK2-dependent cytokines have a central role in tumor immunosurveillance (type I IFN, IL-12, see Figure 1), some are thought to be important based on assigned functions in immunity or inflammation (e.g., IL-23, IL-10, IL-22) and some are attracting attention for evolutionary reasons (e.g., IL-26, type III IFN; [113,114]). Depletion of TYK2 does not completely abrogate tyrosine phosphorylation of downstream STATs by all three families of cytokine receptor (Table 1 and see below), suggesting that TYK2 is the subordinate JAK and mainly amplifies/sustains responses [61]. The partial functionality of the receptors in the absence of TYK2 is consistent with reports that high doses of cytokines (e.g., type I IFN, IL-12, IL-23, IL-10, IL-22) overcome the phenotypic consequences of TYK2 deficiency in primary cells and *in vivo* (e.g., [66,85,95,96,97]).

### 4.1. Type I IFNs

The type I IFN cascade is one of the most extensively studied signaling pathways. In men and mice, type I IFNs belong to a multigene family containing numerous IFNα subtypes and one IFNβ, IFNε, IFNκ, IFNω (absent in mice) and IFNζ (also called limitin, absent in men) [117,118,119]. As subtype-specific functions in antiviral and antiproliferative activities are well described [120,121,122], we focus here on the activity of the type I IFN family in the context of anticancer immunity.

Under homeostatic conditions, most cells produce moderate amounts of type I IFN, but specialized cells of the immune system, mainly plasmacytoid dendritic cells (pDCs), produce high amounts when activated [123,124,125]. All type I IFNs signal through a ubiquitously expressed receptor composed of the IFNAR1 and the IFNAR2 chain (Table 1). When the corresponding cytokine binds to the high-affinity IFNAR2, the TYK2-associated IFNAR1 is recruited to form a ternary signaling-competent complex that activates JAK1 and TYK2 [122,126]. The major signal is through STAT1/STAT2 dimers that, together with interferon regulatory factor 9 (IRF9), form transcription factor IFN-stimulated gene factor 3 (ISGF3) and bind to IFN-stimulated response elements (ISRE) in regulatory regions of IFN-stimulated genes (ISGs). STAT1 homodimers are less strongly activated and bind to IFNγ-activated sequences (GAS) [47]. In macrophages, type I IFN converts the ISGF3 subunits from a STAT2-IRF9 complex that drives homeostatic expression to the complete STAT1-STAT2-IRF9 complex [127], thereby inducing the transcription of an extended set of genes. In addition, non-canonical STAT-containing complexes, such as unphosphorylated ISGF3, ISGF3-II (which consists of STAT1, unphosphorylated STAT2 and IRF9) and STAT2/IRF9, contribute to type I IFN responses at distinct stages after signal initiation [128,129,130,131,132]. The response to type I IFN is made even more plastic by cell-type specificities at multiple steps of the signaling cascade, including the activation of STAT3, STAT4, STAT5 and STAT6 [122,133]. 

Over a thousand ISGs have been described [134]. Most of them encode proteins with antiviral, immunomodulatory and/or (auto-)inflammatory functions [135,136,137,138]. Immunomodulatory and anticancer functions include the direct induction of cytokines (e.g., IFNγ, IL-12, IL-15) and chemokines or the expansion of cytokine-producing cells (e.g., IFNγ-producing T cells), the maturation and activation of DCs and natural killer (NK) cells, proliferative and anti-apoptotic effects on effector T cells and the inhibition of T-regulatory (Treg) or myeloid-derived suppressor cells (MDSC) [139,140,141,142]. The reduced number of Treg cells and their functional suppression has been demonstrated with Ifnar1*^−^*^/−^ mice or by delivery of type I IFN into the TME. It is context-dependent and is attributed to crosstalk with antigen-presenting cells (APCs), enhanced IL-6 activity and a suppressive Treg gene signature [143,144,145]. Pattern recognition receptor (PRR) signaling and type I IFN induction counteracts MDSC expansion and function [146,147]. Tumor cell-intrinsic type I IFN functions include the induction of tumor cell death, enhanced MHC class I expression and tumor antigen presentation [139,140,141,142]. However, not all IFN functions are anti-tumorigenic: type I IFN targets the genes encoding various immune checkpoint proteins (e.g., PD-1, PD-L1) [148,149] and sustained exposure to type I IFN leads to T cell exhaustion or resistance to immune checkpoint therapy [150,151].

The full activation of STAT1-3 upon type I IFN stimulation absolutely requires TYK2, as proven in human and murine LOF mutants (see above and Table 2), while the contribution of TYK2 to the activation of STAT4-6 and to the formation of non-canonical STAT complexes remains to be elucidated.

### 4.2. IL-12 and IL-23

The heteromeric IL-12 and IL-23 cytokines contain a common cytokine subunit (p40) and signal through receptor complexes that share the TYK2-associated IL-12Rβ1 receptor chain (Table 1). The specific receptor chains transduce the signal and are associated with JAK2. IL-12Rβ1 is expressed on T cells, NK cells and DCs [152,153,154]. Although similar signaling pathways are activated by IL-12 and IL-23, the cellular responses differ markedly [155,156,157].

#### 4.2.1. IL-12

IL-12 is largely produced by myeloid cells and DCs in response to activation of PRRs by microbial or damage-associated stimuli [158,159]. IL-12 is composed of p35 and p40 and signals through IL-12Rβ1 and IL-12Rβ2. The IL-12-specifc IL-12Rβ2 chain is expressed mainly on activated T cells and NK cells but also on other cell types, such as DCs and, at a low level, on non-activated NK cells [160,161]. It is upregulated in the presence of cytokines, such as IFNα/β, IFNγ, IL-18 and IL-27 [162,163,164,165], and is downregulated when IL-4 is induced [166]. IL-12 activates STAT4, leading to high levels of IFNγ. To a lesser extent it also activates STAT1 and STAT3-5, which explains its pleiotropic functions that include the regulation of many other cytokines, chemokines and receptor chains [161]. IL-12 is central to the coordination of innate and adaptive immunity and has critical roles in Th1 differentiation [153,154,161]. Its anticancer and anti-metastatic properties have been extensively examined in a variety of murine tumor models. They include the sustained production of IFNγ from NK and T cells as well as the stimulation of proliferation and cytotoxicity of activated NK cells and CD8^+^ and CD4^+^ T cells [167,168]. IL-12 appears to be more potent in tumor surveillance when it is in the TME than when it is systemic [169].

The dependence of *in vivo* IL-12 responses on TYK2 is clear in Tyk2*^−^*^/−^ mouse models of infection, inflammation and cancer, in Tyk2^P1124A^ mouse models of inflammation ([61,79,82] and see below) and in patients with inherited LOF TYK2, who suffer from increased microbial infections and skin inflammation ([43,81] and see Table 2).

#### 4.2.2. IL-23

IL-23 is largely produced by APCs upon infectious and inflammatory stimuli [170,171,172]. It is composed of p40 and p19 and signals through IL-12Rβ1 and IL-23R. IL-23R is present on activated and memory T cells, on innate lymphoid cells (ILCs), including NK cells, and, at lower levels, on monocytes/macrophages and DCs [173,174]. Many of these immune cells carry IL-23R constitutively, while others only express the gene at particular stages of differentiation [170]. IL-23 signaling causes the formation of STAT3/STAT3, STAT3/STAT4 and STAT4/STAT4 dimers and, like IL-12, IL-23 also induces the phosphorylation of STAT1 and STAT5. The contributions of the distinct STAT-containing complexes to the IL-23 response are not fully elucidated [175,176]. Although IL-23 does not directly promote Th cell differentiation, it is a key cytokine for the maintenance and activity of Th17 cells. IL-23 induces the expression of pro-inflammatory cytokines, such as IL-17A, IL-17F and IL-22, of the Th17 lineage-defining transcription factor RORγc and of its own specific receptor chain IL-23R [157]. IL-23 also acts on other cell types and, like IL-12, induces the transcription of IFNγ [153,161]. 

IL-23 acts antagonistically to its close relative IL-12 [177]. It is a central molecular link between the Th17-driven pro-inflammatory processes that promote tumor growth and the tumor-inhibiting reduction of the Th1 signature and cytotoxic T lymphocyte (CTL) activity in the TME [178,179]. Its dependence on TYK2 has only been studied in inflammatory conditions and bacterial infections in mice [96,97,180] and in infectious diseases in men [77,181] (and see Table 2).

### 4.3. IL-10 Family Cytokines

The TYK2-associated IL-10R2 (also IL-10Rβ, Table 1) is shared by a diverse set of cytokines, namely IL-10, IL-22, IL-26, and the more distantly related IFNλ (type III IFN, also IL-28A, IL-28B, IL-29). The second and specific receptor chains bind with high affinity and transduce the signal, and all are associated with JAK1. IL-10R2 is relatively widely expressed, and its cell-type specificity is mainly determined by the second receptor chain [182]. 

#### 4.3.1. IL-10

T cells are the major source of IL-10, but other immune cells, including APCs, NK cells, and B cells, also produce it [183,184,185]. IL-10 production in myeloid cells is largely driven by signals downstream of various PRRs, while IL-10 production in T cells is driven by lineage-specifying cytokines (e.g., IL-12, IL-4, and TGFβ), although it is enhanced by a variety of other cytokines [186]. Expression of IL-10R1 (also IL-10Rα) is restricted to leukocytes and is particularly high in monocytes and macrophages. Binding of IL-10 to its receptor leads to the JAK1/TYK2-dependent phosphorylation of STAT3, which is the major signal transducer for IL-10. STAT1 and STAT5 are also activated, but their contribution to the IL-10 response remains unclear. IL-10 is an irreplaceable anti-inflammatory cytokine that acts mainly on macrophages and DCs to limit excessive inflammation and to inhibit cytokine production by Th1 cells [187,188,189]. IL-10 also exerts immunostimulatory and pro-inflammatory effects, for example, enhancing B-cell, granulocyte and mast-cell differentiation, and NK-/T-cell cytotoxicity [190,191]. It is becoming increasingly evident that IL-10 has a dual role in tumorigenesis. It contributes to the initial immune response to cancer by substantially boosting the proliferation and cytolytic activity of CD8^+^ T cells in combination with IL-2 and in combination with IL-18 by increasing the frequency and cytolytic activity of NK cells [192,193]. However, prolonged production of IL-10 in the TME is a potent promoter of cancer as it contributes to the constitutive activation of STAT3 in immune cells and tumor cells, thereby restraining the anti-cancer immune responses and driving oncogenic signaling [194].

The role of murine TYK2 in IL-10 signaling is unclear. It may depend on the cell type or the cellular activation state [61,195]. Human TYK2-deficient patients show impaired IL-10 responses [43], while TYK2^P1104^ patients show unperturbed IL-10 signaling [77]. That the enzymatic activity of TYK2 in IL-10 signaling in primary human (as well as murine) cells is at least partially redundant is evident from experiments with TYK2-selective inhibitors: JAK1 inhibitors strongly blunt IL-10 responses [109].

#### 4.3.2. IL-22

IL-22 is produced by many types of immune cells, including αβ and γδ T cells and ILCs in lymphoid tissues; and macrophages, neutrophils, DCs, and fibroblasts in non-lymphoid tissues. The non-lymphoid sources produce less IL-22 than the lymphoid sources [196,197]. IL-22 acts through high affinity binding to the JAK1-associated IL-22R1, which is only present on epithelial cells, keratinocytes, and hepatocytes. IL-22 mainly signals through STAT3 but also activates low levels of STAT1 and STAT5 [198,199]. IL-22 has important functions in tissue remodeling and repair during inflammation and in antimicrobial defense [196,200,201,202]. It confers pro-survival, cell migratory, and angiogenic capacity that—at least initially—protects against tumor formation, although the same functions support tumor growth and metastasis at later stages of cancer. IL-22 promotes tumors of epithelial origin, such as lung, liver, pancreatic, and gastrointestinal cancers [203,204,205].

Experiments in LOF mice have shown that TYK2 is required for IL-22 responses during skin and intestinal inflammation [95,96]. The only evidence for the role of the IL-22-TYK2 axis in men comes from the pharmacological inhibition of TYK2, which blocks IL-22 responses in various human cells [108,109].

#### 4.3.3. IL-26

Most vertebrates have IL-26, although it is not found in rats or mice. It is secreted by immune cells, including Th17 cells, NK cells, and macrophages, as well as by fibroblasts [206]. It binds to IL-20R1 with high affinity and requires IL-10R2 to form a receptor complex capable of transducing a signal [207,208]. IL-20R1 is present on a variety of non-hematopoietic tissues and cells and on some hematopoietic cells, such as neutrophils and primary blood monocytes [58,209]. IL-26 activates STAT3 and to a lesser extent STAT1 [207,209]. It is not yet been shown to activate JAK1 and TYK2 in human cells but can be assumed to do so as these factors are associated with IL-20R1 and IL-10Rβ2, respectively. In addition to binding to the conventional cytokine receptor, IL-26 is a carrier molecule for DNA and activates PRRs [210]. IL-26 was discovered relatively recently, and its cellular responses are still poorly known. Its major biological functions seem to be linking T-cell functions to epithelial cells and in antimicrobial and inflammatory activities [206,210,211]. As it is not found in rodents, most of its (patho-)physiological properties have been described *ex vivo* using human or chicken cells. There is only one report of its involvement in tumorigenesis, which shows that it activates STAT3 in gastric cancer [212]. It is likely that the extracellular DNA carrier capacity of IL-26 and/or its aberrant expression may drive the chronicity of inflammation and hence tumor progression. The first report of the IL-26-triggered phosphorylation of TYK2 stems from chicken cells, in which TYK2 seems to interact with JAK2 rather than with JAK1 [213].

#### 4.3.4. Type III IFNs

Four type III IFN (IFNλ) members are known in men (IFNλ1-3, formerly IL-29, IL-28A, and IL-28B, and the recently identified IFNλ4) and two in mice (IFNλ2 and IFNλ3) [59]. As is the case for tape I IFN, type III IFN can be produced by all cells upon activation of a variety of PRRs. The mode of induction and regulation is different in myeloid and epithelial cells [214,215]. All IFNλs signal through the TYK2-associated IL-10R2 and the JAK1-associated IL-28R (also called IFNλR1). IL-28R is mainly restricted to epithelial cells and human, but not murine, hepatocytes. Some subsets of macrophages and DCs in men and mice and NK cells and neutrophils in mice (controversially for men) have also been reported to respond to IFNλ [60,216,217]. Similar to type I IFNs, IFNλs mainly induce the phosphorylation of STAT1/STAT2 heterodimers, which interact with IRF9 to form the ISGF3 complex, although STAT3-6 is also activated [217]. In common with type I IFN, although in fewer cell types, IFNλ shape immunity to infection and cancer and can elicit autoimmune reactions. The current model is that type III IFN acts as the initial line of defense against infection and/or damage to epithelial barriers, while the more potent type I IFN response is activated when local responses are insufficient [60,117,217,218,219].

As they are not active in all cell types, type III IFNs are less involved in the systemic over-stimulation of the immune response or exacerbation of inflammation, suggesting that they might usefully be applied in cancer therapy [220,221]. Particular attention is being paid to IFNλ-activated pDCs that produce type I IFNs and chemokines, thereby recruiting activated T lymphocytes and orchestrating complex immune cell interactions [222]. Analogous to type I IFN, type III IFN targets genes that promote tumorigenesis, depending on context and tumor stage [220].

Unperturbed or suboptimal IFNλ signal transduction is found in TYK2-deficient patient cells [42,43], and pharmacological inhibition of TYK2 reduces the IFNλ responses [107,108].

## 5. Cytokines Produced in Major Dependence of TYK2

Cytokine signaling is highly integrated and depends on crosstalk as well as feedforward and feedback mechanisms. Cytokine responses generally lead to altered patterns of cytokine production. We shall confine ourselves to a brief review of the cytokines that are produced in TYK2 responses (Figure 1).

### 5.1. Type I IFN

Type I (and III) IFN expression is upregulated by IRF3, IRF5, IRF7, and IRF8 downstream of PRRs. As IRF7 is a type I IFN response gene, its expression results in a feedforward loop that drives maximum expression of type I IFN [223,224]. We and others have shown that the tonic and the type I IFN-inducible level of IRF7 transcript drops dramatically in the absence of TYK2 in human and murine cells [225,226]. In addition, systemic IFNβ production in an endotoxemia model is severely reduced in *Tyk2*-deficient mice [99].

### 5.2. IL-15

IL-15 transcription is highly induced by type I IFN and PRR signals in a variety of tissues and cell types. Complex post-transcriptional regulation largely limits IL-15 production to monocytes, macrophages, and DCs. The IL15 receptor belongs to the common gamma chain (γ_C_, CD132) receptor family. The interaction of IL-15 with cells is unique. Not only does it signal in a soluble form, it also binds to IL-15Rα on the cell surface and is trans-presented to cells that express the IL-2/IL-15Rβ and γ_C_ receptors. The IL-15/IL-15Rα complex can also be cleaved into the extracellular space [227,228]. IL-15 is central to the development, survival, and activation of APCs, mast cells, neutrophils, NK cells, T cells, and B cells and enhances NK and CD8^+^ T-cell cytotoxicity. IL-15 is dynamically regulated in the TME and enhances anti-tumor immunity [227,229,230]. We have shown that the absence of TYK2 in DCs results in reduced levels of IL-15Rα on DCs and macrophages in naïve mice ([93] and see below). However, the nature of the TYK2-dependent signals that drive IL-15/IL-15Rα expression under homeostatic conditions is unclear.

### 5.3. Type II IFN

Type II IFN (IFNγ) is primarily produced by cells of the innate and adaptive immune system, including NK cells and innate lymphoid cells (ILCs), T helper 1 (Th1) cells, and CD8^+^ CTLs. IFNγ is the most potent mediator of IL-12, but its production is also induced by other cytokines (primarily IL-18 and IL-23) or through the activation of PRRs or reactive antigen receptors. The IFNGR complex is widely expressed and, similar to IFNγ, is subject to a positive feedback loop during T cell differentiation. IFNγ shapes the TME by enhancing the activity of macrophages, CTLs, and Th1 cells, as well as by suppressing Treg function, angiogenesis, and metastasis. It also acts directly on tumor cells, increasing their antigenicity and promoting cell death. However, like most other cytokines, IFNγ induces inhibitory feedback mechanisms (e.g., upregulating SOCS proteins [53,231]), inhibitory receptor signaling (e.g., PD-L1, [232]) and other genes involved in immunosuppression or immune-evasion [233,234,235]. Mutations in IFNγ pathway genes underlie resistance to immune checkpoint therapies, i.e., block of PD1/PD-L1 or CTL4-A [236,237,238]. We and others have established that IL-12- or IL-23-activated TYK2 is central to the production of IFNγ during immunity to tumors and to infections as well as during inflammation ([52,61,93] and see Table 2).

### 5.4. IL-17

The IL-17 family consists of IL-17A-F and is crucial for the host defense against microbial infections and the development of inflammatory diseases. Although IL-17A is the signature cytokine produced by T helper 17 (Th17) cells, IL-17A, and other IL-17 family cytokines are produced by many immune and non-immune cells [239,240]. The IL-23-TYK2 axis is central to the maintenance of Th17 cells and sustained IL-17 production. In many human malignancies, a Th17 cell signature (*RORC*, *IL17*, *IL23*, *STAT3*) is associated with poor outcome, whereas Th1 cell signatures (*IFNG*, *STAT1*, *TBX21*) and CD8^+^ CTLs (*PRF1*, *GZMB*) are linked to better overall survival [240,241,242,243]. Th17 cells and associated cytokines in the TME can promote tumorigenesis by perpetuating inflammation and/or aggravating the aggressiveness of the tumor cells. The importance of TYK2 in the regulation of IL-17A production in men and mice is well documented [43,77,91,96,97,103,244,245,246].

## 6. TYK2 and Immunity to Cancer

Human TYK2 LOF patients have not been reported to be more prone to tumors, although this surprising non-finding probably relates largely to the patients’ ages and treatment histories. In cancer databases, LOF or lowered TYK2 levels are generally correlated with poor prognosis, while high TYK2 levels are associated with tumorigenesis. The discrepancy may relate to the lack of information on whether TYK2 is in the tumor or the TME and/or to the lack of distinction between phosphorylated and unphosphorylated TYK2 [247].

The first link between TYK2 and immunity to cancer came from studies in *Tyk2*-deficient mice, which develop hematopoietic malignancies upon oncogenic induction or systemic transplantation of tumors, for example, Abelson-induced B-lymphoid leukemia/lymphoma, Tel-JAK2-induced T-lymphoid leukemia, and EL4 lymphoma [86,98]. The high susceptibility of *Tyk2**^−^*^/−^ mice to lymphoid tumors is the result of an impaired tumor surveillance, in particular, a decreased cytotoxic capacity of NK/NKT and CTL cells [86,98]. The importance of TYK2 for NK cell- and CD8^+^ T cell-targeted tumor growth restriction was confirmed by studies with local tumor transplant models, using lymphoma (RMA-S and RMA-Rae1) and adenocarcinoma (MC38) cells [67,76,93]. Reduced CTL-dependent tumor cell killing is linked to impaired type I IFN signaling, although the mechanisms are elusive [98]. Defective NK cell-dependent tumor growth control in the absence of TYK2 correlates with defects in splenic NK-cell maturation; NK-cell development in the bone marrow is independent of TYK2 [76]. Studies with conditional knockout mice have shown that TYK2 in DCs (CD11c^+^) and other non-NK cells, rather than in NK cells themselves, drives terminal NK cell maturation in the spleen [93]. The work also implicates TYK2 in the regulation of IL-15Rα on CD11c^+^ cells but not NK cells or T cells. Administration of exogenous IL-15/IL-15Rα to *Tyk2*-deficient mice restores NK-cell maturation and NK cell-dependent control of tumor growth [93]. In contrast to the situation in NK cells, the CTL-dependent restriction of locally transplanted tumors is independent of TYK2 in myeloid cells and DCs [67]. TYK2 has a kinase-independent function in NK cell-mediated tumor surveillance, as evidenced by the partial restoration of NK-cell maturation and cytotoxicity in *Tyk2^K923E^* mice [76].

Work in a murine model of breast cancer has shown that TYK2 deficiency enhances tumor growth and metastasis by stimulating MDSCs rather than by inhibiting the activity of T cells or NK cells [87]. TYK2 signaling has also been implicated in tumor-cell infiltration and metastasis in human prostate cancer patients [248], a finding recapitulated in Eμ-Myc transgenic mice, in which TYK2 deficiency reduces tumor-cell invasiveness into the liver [249]. For further details on the role of TYK2 in tumor-cell invasiveness, we refer the reader to a recent review [247].

The data from mouse models indicate that the functions of TYK2 in tumor surveillance are context-dependent. Although TYK2-dependent cytokines have been implicated in a plethora of processes in anti-tumor immunity (Figure 1), the consequences of TYK2 deficiency on tumor immunity are only beginning to emerge. The fascinating challenge to deciphering the impact of TYK2 in inhibiting or promoting cancer growth stems from its amplifier function, which exacerbates the general complexity of the spatial-temporal activities and networked hierarchies in signaling circuits. The contribution of TYK2 to the TME is hard to predict as cell-type specific and local high concentrations of cytokines might overcome or mask the effects of TYK2 deficiency.

## 7. Conclusions and Outlook

The pivotal role of TYK2 in (auto-)immune and -inflammatory diseases was established in gene-targeted mice and human patients with mutated *Tyk2*/*TYK2* alleles. Studies of TYK2 in anticancer immunity in mice have revealed a pivotal role of TYK2 in the NK- and T-cell-mediated elimination of tumor cells. The conditional ablation of TYK2 uncovered the cell-intrinsic and -extrinsic requirements for TYK2 in the crosstalk between DCs, macrophages, and cytolytic effector cells. As the other JAK family members, TYK2 also has oncogenic properties: these are reviewed elsewhere [52,247,250].

The pharmacological inhibition of TYK2 (TYK2inibs) is a recognized approach to the treatment of chronic inflammatory and autoimmune diseases and is under consideration for cancer treatment. Several TYK2inibs are in clinical trials [108,251,252,253]. A CTLA-4-TYK2-STAT3 axis has been reported in B cell lymphoma cells and tumor-associated B cells [254] and will be relevant to immune checkpoint therapy. A TYK2inib has promising immunotherapeutic effects in several tumor transplant mouse models. The TME had reduced levels of PD-1, indicative of less immune cell exhaustion and reduced Treg infiltration [255]. The translation of these findings into the clinic would represent an important extension of the use of TYK2inibs in cancer treatment.

We are applying advanced sequencing methods to obtain detailed mechanistic insights into the crosstalk and the kinase-dependent functions of TYK2. We are creating an integrated map of the chromatin landscape and the activities of splenic myeloid cells, NK cells, DCs, and T cells in homeostatic conditions and intend to apply the same strategy to the TME and/or splenocytes under tumor burden. This will enable us to define the molecular/functional intersection of TYK2 as a tumor surveyor and/or as a promoter of carcinogenesis by shaping immune and oncogenic pathways. Complementary to analyzing the chromatin architecture, it would be highly informative to determine the cellular interactomes of TYK2 in the TME by quantitative proteomics [256], single cell proteomics [257,258], or multidimensional imaging [259]. This would also pave the way for determining the (patho-)physiological relevance of the subcellular distributions of JAKs and STATs, for example, revealing the nuclear functions of TYK2 [260] and other JAKs [261] and the impact of TYK2 [262] and STATs on mitochondrial function [263,264]. Our ultimate vision is to use the ‘relative’ molecular simplicity of the JAK-STAT axis as a template for the development of computational pathway modeling and prediction [265,266,267].

## Figures and Tables

**Figure 1 cancers-12-00150-f001:**
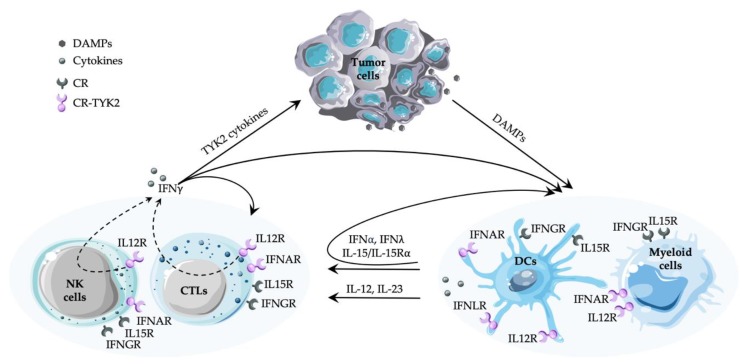
Simplified scheme of the cancer-immunity cycle and the involvement of TYK2, focusing on innate and adaptive immune cells in tumor surveillance and the feedforward loops in the TYK2-dependent cytokine responses and production. Type II IFN and IL-15 production largely depends on TYK2, while the cytokine receptors signal independently of TYK2. Type I and III IFNs, IL-12 and IL-23 act through TYK2-dependent receptors, with an amplification of type I and III IFN production by TYK2 signaling. Effects of the cytokines on shaping the TME and the stroma cells are reviewed elsewhere [19,115,116]. CR, cytokine receptor; CR-TYK2, TYK2-dependent cytokine receptor; DAMPs, danger-associated molecular patterns; IFNα, type I IFN; IFNγ, type II IFN; IFNλ, type III IFN; IFNLR, receptor for type III IFN, IFNλ; IL12R, receptor for family members IL-12 or IL-23; IL15R, simplified for IL-15 responses through IL-15/IL-15 Rα trans-presentation or soluble IL-15 binding to IL-15Rβ/γ_C_/IL-15Rα (see text); TYK2 cytokines, cytokines depending on TYK2 with respect to signaling and/or production. The figure was compiled with Servier Medical Art (https://smart.servier.com).

**Table 1 cancers-12-00150-t001:** TYK2-dependent cytokine receptor signaling validated by loss or inhibition of TYK2.

Cytokine Family	Receptor (R) Chain (⇔ TYK2)	Cytokine & Specific R Chain (⇔ JAK1 or 2)	STATs Activated ^1^	Producing Cells	Responding Cells
**Type I IFNs**	IFNAR1	**IFNα/β**^2^ IFNAR2 ⇔ JAK1	**STAT1**, **STAT2**^3^ STAT3-6	all cells, pDCs (professional type I IFN producers)	all cells
**IL-12/23 family**	IL-12Rβ1	**IL-12** IL-12Rβ2 ⇔ JAK2	**STAT4** STAT1/3/5	monocytes, macrophages, DCs	activated NK and T cells
		**IL-23** IL-23R ⇔ JAK2	**STAT3**, **STAT4** STAT1/5	monocytes, macrophages, DCs	activated T cells, ILCs, NK cells, monocytes, macrophages, DCs
**IL-10 family**	IL-10R2	**IL-10** IL-10R1 ⇔ JAK1	**STAT3** STAT1/5	Monocytes ^4^, macrophages ^4^, DCs NK cells ^5^, T cells, B cells	all subsets of leukocytes
		**IL-22** IL-22R1 ⇔ JAK1	**STAT3** STAT1/5	T cells, ILCs, NK/T cells, macrophages, neutrophils, DC, fibroblasts	epithelial cells, keratinocytes, hepatocytes
		**IL-26**^6^ IL-20R1 ⇔ JAK1	**STAT3** STAT1	Th17 cells, NK cells, macrophages, fibroblasts	non-hematopoietic and hematopoietic cell subsets ^7^
		**type III IFNs/IFNλ** IL-28R ⇔ JAK1	**STAT1**, **STAT2**^3^ STAT3-6	all cells ^8^	epithelial cells, DCs, neutrophils a.o. ^9^

⇔, associated with; a.o., and others; ^1^ bold font, main STATs activated by respective cytokine receptor engagement; normal font, STAT activation is dependent on cell type or of less clear biological relevance; ^2^ see text for complete nomenclature of subtypes; ^3^ trimer of STAT1-STAT2-IRF9 (ISGF3); ^4^ anti-inflammatory activity; ^5^ immune stimulatory activity; ^6^ does not exist in rodents; ^7^ for details see [57,58]; ^8^ overlap with IFN type I production, for differences in IFN type III and I induction see text and [59]; ^9^ for cell type and species-specificity see text and [60]. Gp130-utilizing cytokines (IL-6, IL-11, IL-27, IL-35, CNTF, CT-1, LIF, OSM) can activate TYK2, although there is no evidence that TYK2 is required for the downstream activation of STATs [61]. The dependency of IL-4/IL-13 signaling on TYK2 is understudied in LOF patients and mice; see other reviews [62,63].

**Table 2 cancers-12-00150-t002:** Inherited and gene-targeted mutations of TYK2 in mice and men.

		Mutation	Disease	IL-10	IL-12	IL-22	IL-23	IFN-I	IFN-III
**mouse**	LOF of TYK2	*Tyk2* ^−/−^	impaired tumor surveillance; susceptible to microbial infections; resistance/protection during inflammatory challenges	normal [64,66]; impaired [85]	impaired ^1^ [76,86,87] ^2^; [64,66,85,88,89,90,91,92,93,94]	impaired [95,96]	impaired [91,94,96,97]	impaired [98] ^2^; [64,66,74,99,100,101]	n.d.
		*Tyk2^mut^*	susceptible to parasite infection; protection during inflammatory challenge	n.d.	impaired [68,102]	n.d.	impaired [68,102]	impaired [68,99]	n.d.
		*Tyk2^Pmut^*	virus-induced diabetes	n.d.	n.d.	n.d.	n.d.	impaired [69]	n.d.
	KI-TYK2	*Tyk2^K923E^*; *Tyk2^K923R^*	impaired tumor surveillance; susceptible to virus infection; restoration of obesity	n.d.	impaired [76] ^2^	n.d.	n.d.	impaired [74,75]	n.d.
		*Tyk2^P1124A^*	protection during inflammatory challenge	n.d.	impaired [79,82]	n.d.	impaired [79,82]	impaired [79,82]	n.d.
**human**	LOF of TYK2	patient 1	bacterial, viral and fungal infections, HIES	impaired [72,103]	impaired [72,103]	n.d.	impaired [72]	impaired [72]	n.d.
		patients 2–10	bacterial and viral infections, no HIES ^3^	impaired [42,43,103]	impaired [42,43,73,103]	n.d.	impaired [43]	impaired [42,43]	impaired [43]; normal [42]
		promoter variants	virus-induced type I diabetes	n.d.	n.d.	n.d.	n.d.	impaired [70,71]	n.d.
	KI-TYK2	*TYK2^P1104A^*	mycobacterial infection; autoimmune protected/susceptible	normal [77] ^4^ [79]/[78]	normal [77,78,81] ^5^; impaired [79]	n.d.	impaired [77,79,82] ^6^; normal [78] ^5^	normal [77]; impaired [78,79,82]	n.d.
		*TYK2^I684S^*	healthy; autoimmune protected	normal [77]	normal [77]; impaired [81]	n.d.	normal [77]	normal [77]	n.d.

LOF of TYK2, deficient of protein or substantially lowered protein level; KI-TYK2, kinase-inactive TYK2; *Tyk2^mut^*, natural TYK2 deficient mouse strain; *Tyk2^Pmut^*, Tyk2 promoter mutation; n.d., not determined; ^1^ note that in [86,87] IFN-II production is attributed to impaired IL-12 responses; ^2^ references related to tumor surveillance; ^3^ note that the patient in [42] shows IL-6-independent HIES-like symptoms; ^4^ note that cytokine signaling was analyzed in EBV-transformed patient and healthy donor cells; ^5^ normal Th1 and Th17 polarization indicative of no gross impairment of IL-12 and IL-23 signaling [78]; ^6^ not reaching statistical significance in [82]. IL-6 signaling is unaffected in LOF of TYK2 and KI-TYK2 patients and mice [42,43,64,66,77,79]; IL-13 signaling was not assessed in LOF of TYK2 men and mice and found unaffected in KI-TYK2 patients [79]

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
