# Peer review of "TYK2 in Tumor Immunosurveillance"

_cancers, 2020, doi:10.3390/cancers12010150_

Round 1
Reviewer 1 Report
This review paper, written by expert scientists in the field, provides an updated view on the role of TYK2 kinase in tumor immunosurveillance. The review covers all this topic, carefully providing an exhaustive overview on the role of TYK2 in cancer development.
My only remark is that it is surprising that the authors do not mention studies on the role and on the mechanism of TYK2 activation in anaplastic large cell lymphoma (see Prutsch et al., Leukemia 2019; Crescenzo et al, Cancer Cell 2015).
Author Response
This review paper, written by expert scientists in the field, provides an updated view on the role of TYK2 kinase in tumor immunosurveillance. The review covers all this topic, carefully providing an exhaustive overview on the role of TYK2 in cancer development.
My only remark is that it is surprising that the authors do not mention studies on the role and on the mechanism of TYK2 activation in anaplastic large cell lymphoma (see Prutsch et al., Leukemia 2019; Crescenzo et al, Cancer Cell 2015).
Reply: We thank the reviewer for the interest in the oncogenic role of TYK2; due to the limited space we refer the reader in the Abstract and the first paragraph of the Conclusion to reviews covering this topic.
We apologize for the style of the manuscript. The review has been now thoroughly revised by a native speaker.
Reviewer 2 Report
Karjalainen et al. provide an interesting review, which addresses the functional role of TYK2-dependent cytokine signaling in immune cells in the context of tumor immunosurveillance. Although the review is timely and interesting, it needs to be revised. Some passages are slightly difficult to follow. This review would benefit significantly if vetted by a native English speaker; in its current state it is tedious to comprehend. Specific attention must be paid to spelling errors and to simplify the sentences across the manuscript.
Specific Points:
In Table 1 – other cytokines discussed in the review – IL-6 as well as cytokines such as IL-4/IL-13 are excluded. If there is a particular reason that these cytokine families were not included, then the reason behind this exclusion should be discussed in the review. Table 2 needs to be formatted and organized better. The columns should be formatted such that the Table can be comprehended easily. Font size can be adjusted so that the columns can fit better. In this huge table, the emphasis is on diseases other than cancer and this distracts from the main focus of this review – role of TYK2 and cancer. Why not focus on partially impaired tumor surveillance section of the table? Figure 1: The Figure Legend should include a brief description of the Figure – in its current version, it is very difficult to understand the significance of the Figure and how it relates to the description given in the text. The legend talks about Type II IFN (IFN-g) effects on tumor cells –however, the boxes above the figure talks about many other TYK2-dependent cytokine effects? This is very confusing. Labels are missing for tumor cells, spelling errors in the figure – there is a dollar sign next to IFN. Sections 5 and 6 are the most relevant sections to this review – these can be further expanded and moved ahead of Section 4. In section 4, the sentence – “Note that frequency TYK2 activation is biochemically determined by phosphorylation of the critical tyrosine residues in the activation loop which cannot be put on a level with further dissected downstream cellular effectors” This sentence is confusing and should be clarified. In page 8, second paragraph –“decrease of T regulatory or myeloid-derived suppressor cells (MDSC functions)” - the authors should describe how Type I interferon signaling decrease these immune subsets in the TME and it will be helpful if the appropriate references are cited. Also, in the same paragraph – mutations in Type I IFN response genes also results in anti-PD-1 antibody-mediated therapeutic resistance in metastatic cancer patients – this can be included as well and the references cited. The dependency of IL-23, IL-10, IL-22, IL-26 and Type III IFN cytokines on TYK2, especially in the context of cancer, has not been clearly established – why are these cytokines described/ discussed here?Author Response
Karjalainen et al. provide an interesting review, which addresses the functional role of TYK2-dependent cytokine signaling in immune cells in the context of tumor immunosurveillance. Although the review is timely and interesting, it needs to be revised. Some passages are slightly difficult to follow. This review would benefit significantly if vetted by a native English speaker; in its current state it is tedious to comprehend. Specific attention must be paid to spelling errors and to simplify the sentences across the manuscript.
Reply: We apologize for the style of the manuscript. The review has been now thoroughly revised by a native speaker, Graham Tebb who is now co-author.
Specific Points:
In Table 1 – other cytokines discussed in the review – IL-6 as well as cytokines such as IL-4/IL-13 are excluded. If there is a particular reason that these cytokine families were not included, then the reason behind this exclusion should be discussed in the review.
Reply: Table 1 has been complemented by indicating the reasons for excluding the IL-6 family of cytokines and IL-4/IL-13 from the review. The rationale of the selection of the cytokines discussed in this review is also indicated by the changed heading of Table 1 and rephrased in the first paragraph of chapter 4 (see also below). Table 2 has been complemented in the legend with information concerning IL-6 and IL-13 signaling in mutant TYK2 mice and men (see below)
Table 2 needs to be formatted and organized better. The columns should be formatted such that the Table can be comprehended easily. Font size can be adjusted so that the columns can fit better. In this huge table, the emphasis is on diseases other than cancer and this distracts from the main focus of this review – role of TYK2 and cancer. Why not focus on partially impaired tumor surveillance section of the table?
Reply: We apologize for the over-boarding and unclear formatted Table 2. To our knowledge the Table presents the first comprehensive comparison of the outcome of LOF of TYK2 or kinase-inactive TYK2 in human patients and mouse models. A focus on tumor surveillance would remove all human patient information. With respect of size of the Table we reduced it by fusing lines. In addition the Table is presented in landscape format and carefully edited for spelling errors and font sizes.
In addition, we complemented the legend with information concerning IL-6 and IL-4/IL-13 signaling in mutant TYK2 mice and men (see above).
Figure 1: The Figure Legend should include a brief description of the Figure – in its current version, it is very difficult to understand the significance of the Figure and how it relates to the description given in the text. The legend talks about Type II IFN (IFN-g) effects on tumor cells –however, the boxes above the figure talks about many other TYK2-dependent cytokine effects? This is very confusing. Labels are missing for tumor cells, spelling errors in the figure – there is a dollar sign next to IFN.
Reply: We changed the title of the Figure in order to clarify its significance. We removed confusing information and extended the legend accordingly. The missing labels and explanations of symbols and abbreviations are now included.
Sections 5 and 6 are the most relevant sections to this review – these can be further expanded and moved ahead of Section 4.
Reply: We tried to shift Section 4 above and felt that with respect to understanding the flow of information during signaling, i.e. cytokine receptor activation => cytokine response => cytokine production would be not reflected appropriately. Therefore we would appreciate if very much if the reviewer would kindly agree with the given order of chapters.
In section 4, the sentence – “Note that frequency TYK2 activation is biochemically determined by phosphorylation of the critical tyrosine residues in the activation loop which cannot be put on a level with further dissected downstream cellular effectors” This sentence is confusing and should be clarified.
Reply: we are very sorry for the confusion; the sentence was deleted and the rationale for the selection of the cytokines reviewed was rephrased (see also above and below).
In page 8, second paragraph –“decrease of T regulatory or myeloid-derived suppressor cells (MDSC functions)” - the authors should describe how Type I interferon signaling decrease these immune subsets in the TME and it will be helpful if the appropriate references are cited. Also, in the same paragraph – mutations in Type I IFN response genes also results in anti-PD-1 antibody-mediated therapeutic resistance in metastatic cancer patients – this can be included as well and the references cited.
Reply: We thank the reviewer for the suggestion; we have supplemented the information on IFN-I and Tregs and MDSCs and added according references.
We also thank the reviewer for suggesting to comment on resistance to immune checkpoint therapy through mutation in IFN pathway genes. Since these mutations mostly concern the type II IFN pathway we included the information in section 5.3.
The dependency of IL-23, IL-10, IL-22, IL-26 and Type III IFN cytokines on TYK2, especially in the context of cancer, has not been clearly established – why are these cytokines described/ discussed here?
Reply: The rationale of the selection of the cytokines discussed in this review is indicated by the changed heading of Table 1 and rephrased in the first paragraph of chapter 4 (see also above)
Reviewer 3 Report
Several points are treated in this review, for this reasons I suggest only minor revision. In particular, I suggest to Authors report the information/relation for immune checkpoint (e.g. PD-1 and CTLA4) and Tyk2 in the tumor microenvironment. For example, CTLA4-CD86 ligation recruits and activates the Tyk2, resulting in STAT3 activation and expression of genes critical for cancer immunosuppression and tumor growth and survival (Herrmann A et al., Cancer Res 2017.). This question would be interesting, in particular for the tyk2, that this molecule present an important role for " Tumor Immunosurveillance " in the tumor microenvironment. Can the authors comment on the effect of pharmaceutical inhibitors of Tyk2 on immune cells?
Author Response
revision. In particular, I suggest to Authors report the information/relation for immune checkpoint (e.g. PD-1 and CTLA4) and Tyk2 in the tumor microenvironment. For example, CTLA4-CD86 ligation recruits and activates the Tyk2, resulting in STAT3 activation and expression of genes critical for cancer immunosuppression and tumor growth and survival (Herrmann A et al., Cancer Res 2017.). This question would be interesting, in particular for the tyk2, that this molecule present an important role for " Tumor Immunosurveillance " in the tumor microenvironment.
Reply: thank you for the comment; respective information was included in Section 4.1; 5.3 and the Conclusions
Can the authors comment on the effect of pharmaceutical inhibitors of Tyk2 on immune cells?
Reply: thank you for the interest in and reminder for this important issue: see section 4, 1st paragraph and additional paragraph in the Conclusions
Round 2
Reviewer 2 Report
This review has been significantly improved and now warrants publication in Cancers.